# Sickness absence after carpal tunnel release: a multicentre prospective cohort study

Lisa Newington ,[1,2] Georgia Ntani ,[1] David Warwick ,[3] Jo Adams ,[4,5] Karen Walker-Bone [1]

[1]MRC Versus Arthritis Centre for Musculoskeletal Health and Work, MRC Lifecourse Epidemiology Unit, University of Southampton, Southampton, UK
[2]Hand Therapy, Guy's and St Thomas' NHS Foundation Trust, London, UK
[3]Hand Surgery, University Hospital Southampton NHS Foundation Trust and Faculty of Medicine, University of Southampton, Southampton, UK
[4]Centre for Sport, Exercise and Osteoarthritis Research Versus Arthritis, University of Southampton, Southampton, UK
[5]Centre for Innovation and Leadership in Health Sciences, School of Health Sciences, Faculty of Environment and Life Sciences, University of Southampton, Southampton, UK

**Correspondence to**
Dr Lisa Newington;
l.newington@nhs.net

## ABSTRACT

**Objectives** To describe when patients return to different types of work after elective carpal tunnel release (CTR) surgery and identify the factors associated with the duration of sickness absence.

**Design** Multicentre prospective observational cohort study.

**Setting and participants** Participants were recruited preoperatively from 16 UK centres and clinical, occupational and demographic information were collected. Participants completed a weekly diary and questionnaires at four and 12 weeks postoperatively.

**Outcomes** The main outcome was duration of work absence from date of surgery to date of first return to work.

**Results** 254 participants were enrolled in the study and 201 provided the follow-up data. Median duration of sickness absence was 20 days (range 1–99). Earlier return to work was associated with having surgery in primary care and a self-reported work role involving more than 4 hours of daily computer use. Being female and entitlement to more than a month of paid sick leave were both associated with longer work absences. The duration of work absence was strongly associated with the expected duration of leave, as reported by participants before surgery. Earlier return to work was not associated with poorer clinical outcomes reported 12 weeks after CTR.

**Conclusions** There was wide variation in the duration of work absence after CTR across all occupational categories. A combination of occupational, demographic and clinical factors was associated with the duration of work absence, illustrating the complexity of return to work decision making. However, preoperative expectations were strongly associated with the actual duration of leave. We found no evidence that earlier return to work was harmful. Clear, consistent advice from clinicians preoperatively setting expectations of a prompt return to work could reduce unnecessary sickness absence after CTR. To enable this, clinicians need evidence-informed guidance about appropriate timescales for the safe return to different types of work.

## INTRODUCTION

Carpal tunnel syndrome (CTS) occurs when the median nerve becomes compressed within the carpal tunnel at the wrist. Typical

## Strengths and limitations of this study

► This multicentre prospective study, with a large sample size, provides robust evidence to understand return to work issues after carpal tunnel release surgery.

► Participants were recruited from primary care, secondary care and private practice sites, representing the breath of locations where carpal tunnel release is performed in the UK.

► Work absence was the primary outcome and a clear definition was used for its duration with data collected contemporaneously to limit recall bias.

► A standardised method was used to categorise occupations and measure occupational exposures, although this relied on job title, which may not be a true reflection of actual occupational activity.

► All participants underwent open carpal tunnel release, however, the method of carpal tunnel syndrome diagnosis was not independently verified and different case definitions for carpal tunnel syndrome may have been included.

sensory symptoms include pain, paraesthesia and/or numbness in the thumb, index, middle and radial half of the ring finger; motor symptoms include progressive wasting of the thenar muscles. CTS is often associated with marked functional difficulty[1] and treatment is targeted at reducing the median nerve compression by splinting, corticosteroid injection or carpal tunnel release (CTR) surgery.[2 3]

Recent estimates suggest that more than 90 000 CTR procedures will be performed annually in the English National Health Service (NHS) by 2025,[4] and as the peak incidence for CTS falls within the working lifetime,[5] many of these patients will need to return to work after their CTR. However, there is currently no evidence on which to base recommendations about when it might be safe to return to functional activities, including work, after CTR. Our previous survey of UK hand surgeons, primary care surgeons and hand therapists

found that clinicians recommended a wide range of times to return to three specified job roles after CTR: 0–30 days for desk-based work (eg, keyboard, mouse, writing and telephone); 1–56 days for repetitive light manual work (driving, delivery, stacking) and 1–90 days for heavy manual work (eg, construction).[6] However, there has not previously been a prospective study of CTR patients in the UK in which time to return to work was the primary outcome. Therefore, it is not known when UK patients return to different occupational activities after CTR or what influences the duration of work absence. It is also unclear whether earlier return to work has a detrimental effect on postoperative clinical outcomes. Possible consequences of returning to work too soon after CTR include wound dehiscence, infection and delayed healing. Conversely, delayed return to work may increase the risk of progression to long-term sick leave[7] and produce a financial burden for the individual, employer or state.

A systematic review of the prognostic factors associated with return to work after CTR identified 11 studies which evaluated more than 90 potential prognostic factors.[8] Longer durations of work absence after CTR were found to be associated with: receipt of workers' compensation[9]; manual work[10–12]; longer expected durations of work absence[10]; being on sick leave before CTR surgery[13]; self-blame for the hand problem[13] and beliefs that the symptoms were caused by work.[12]

Much of the existing research has been conducted at single sites and involved small numbers of participants. Furthermore, very few studies have specifically explored the influence of a range of occupational factors. The current multicentre prospective cohort study was designed to explore when patients returned to different types of work after CTR and the demographic, clinical and occupational factors associated with duration of work absence. We also investigated whether earlier return to work was associated with poorer clinical outcomes assessed at 12 weeks after CTR.

## METHODS

This was a prospective cohort study and a convenience sample of participants were recruited from 16 sites in England and Wales between March 2017 and August 2018. Recruitment took place before CTR surgery either at the time of listing for surgery, during preoperative screening, or on the day of surgery. At each site, the patient CTR pathway and treatment was carried out as usual. Sites were NHS secondary care (hospital setting), NHS primary care and private hand surgery facilities, representing the range of UK healthcare facilities where CTR is performed. Provision of CTR in the UK was explored through discussion with relevant national organisations (British Society for Surgery of the Hand and Association for Surgeons in Primary Care). Sites were recruited through National Institute for Health Research infrastructure.

Eligibility criteria are shown in box 1. Baseline demographic, general health and occupational information

---

**Box 1   Study eligibility criteria**

**Self-selected by potential participants**
1. Aged over 18 and referred for carpal tunnel release surgery.
2. Routinely work in paid employment for at least 20 hours per week.
3. Plan to return to work after carpal tunnel release surgery.
4. Have not previously had carpal tunnel release surgery on either hand.
5. Have not previously had a serious injury to the same wrist/hand.

**Assessed by recruiting clinician**
1. No planned surgical procedures for conditions other than carpal tunnel syndrome.

---

were collected via self-completed questionnaire at the time of recruitment. The questionnaire also included standardised measures of CTS symptoms[14–16] and hand function.[17] Questionnaire content was informed by the clinical, demographic and occupational factors previously identified in a systematic review of prognostic factors for return to work after CTR,[8] and developed in collaboration with our patient advisory group. The reasoning for item inclusion is provided in online supplemental material 1.

Follow-up questionnaires were completed four and 12 weeks after CTR and collected information about return to work, work functioning, scar symptoms, CTS symptoms and hand function. Study questionnaires are provided as online supplemental materials 2 and 3. Participants were also asked to complete a short weekly diary from the day after surgery until return to work, detailing whether they had returned to work that week, and if so, the date of return. Steps were taken to minimise lost to follow-up after recruitment. To maximise retention, we incentivised with a shopping voucher on completion of the study (£10) and sent up to three reminders using a combination of post, email and text.

Surgical information was collected from the medical records by a member of the participant's clinical team. This included: date of CTR, operated hand(s), nature of anaesthetic, incision size, additional procedures, unexpected findings and suture material. Date, side of CTR and anaesthetic (general/local) were also reported by participants for cross-checking.

### Public and patient involvement

This research was supported by a patient advisory group consisting of six individuals who had previously undergone CTR at different UK sites. Study questionnaires were developed in collaboration with the patient advisors and these individuals also provided their feedback on the preliminary findings.

### Statistical methods

Comparisons were made between those who dropped out of the study before providing any follow-up data and those in the final study sample using prespecified demographic, clinical and occupational variables (table 1).

**Table 1** Participant demographics assessed at baseline in comparison with those lost to follow-up

| | Study participants n=201 (%) | Lost to follow-up n=53 (%) |
|---|---|---|
| Mean age in years [SD] | 52.0 [9.16] | 44.4 [9.55] |
| Sex | | |
| Male | 76 (37.8) | 20 (37.7) |
| Female | 125 (62.2) | 33 (62.3) |
| Body mass index (kg/m$^2$) | | |
| Normal (18.5–24.9) | 48 (23.9) | 9 (17.0) |
| Overweight (25–29.9) | 66 (32.8) | 16 (30.2) |
| Obese (≥30) | 73 (36.3) | 22 (41.5) |
| Smoking status | | |
| Never smoked | 109 (54.2) | 26 (49.1) |
| Current/ex-smoker | 90 (44.8) | 27 (50.9) |
| General health | | |
| Excellent/very good/good | 174 (86.6) | 42 (79.3) |
| Fair/poor | 26 (12.9) | 11 (20.8) |
| Number of comorbidities | | |
| None | 54 (26.9) | 21 (39.6) |
| 1 | 70 (34.8) | 13 (24.5) |
| 2 or more | 77 (38.3) | 19 (35.9) |
| Number of disabling comorbidities | | |
| None | 138 (68.7) | 35 (66.0) |
| 1 | 41 (20.4) | 9 (17.0) |
| 2 or more | 22 (11.0) | 9 (17.0) |
| Mean SF-36 mental health score [SD] * | 65.6 [17.20] | 60.3 [20.41] |
| Mean bilateral CTS-6 score [SD] † | 2.8 [0.77] | 3.0 [0.73] |
| Mean MHQ bilateral activities of daily living score [SD] ‡ | 68.8 [23.64] | 55.7 [28.62] |
| Mean MHQ work function score [SD] ‡ | 66.1 [22.26] | 60.6 [22.61] |
| Type of job contract | | |
| Employed (permanent contract) | 164 (81.6) | 37 (69.8) |
| Self-employed | 31 (15.4) | 13 (24.5) |
| Employed (temporary or 0 hours contract) | 5 (2.5) | 3 (5.7) |
| Type of work§ | | |
| Manual | 77 (39) | 31 (58) |
| Non-manual | 123 (61) | 22 (42) |
| Median level of job demand on hands/wrists [IQR] ¶ | 9 [7–10] | 10 [7–10] |
| Job satisfaction | | |
| Very satisfied | 87 (43.3) | 24 (45.3) |
| Satisfied/fairly satisfied | 92 (45.8) | 24 (45.3) |

Continued

**Table 1** Continued

| | Study participants n=201 (%) | Lost to follow-up n=53 (%) |
|---|---|---|
| Dissatisfied/very dissatisfied | 20 (10.0) | 5 (9.4) |
| Median expected work absence in days [IQR] | 14 [7–28] | 14 [5–21] |
| Expected availability of sick pay | | |
| ≤1 month | 50 (24.9) | 21 (39.6) |
| >1 month | 94 (46.8) | 11 (20.8) |
| Unsure | 57 (28.4) | 21 (39.6) |
| Study site** | | |
| NHS primary care | 73 (36.3) | 13 (24.5) |
| NHS secondary care | 101 (50.3) | 32 (60.4) |
| Private hand surgery facilities | 27 (13.4) | 8 (15.1) |

*SF-36 mental health score ranges from 0 to 100, higher scores indicate better mental health.[35]
†CTS-6 symptom score ranges from 1 to 5, higher scores indicate more severe symptoms.[14]
‡MHQ ranges from 0 to 100, higher scores indicate better functioning.[17]
§Classified using the Office for National Statistics Standard Occupational Classification 2010.[18 19]
¶Job demand scale range 0–10, 10 indicating very demanding on hands/wrists.[13]
**Location where the carpal tunnel release surgery was performed. Surgery in primary care was performed by general practitioners who had completed additional training.
CTS, carpal tunnel syndrome; IQR, Interquartile range ; MHQ, Michigan Hand Questionnaire ; NHS, National Health Service; SF-36, 36-Item Short Form Health Survey .

Manual and non-manual work was coded from job title and industry using the UK Standard Occupational Classification.[18 19] Return to work time was calculated from the date of surgery to the date of first return to work (as reported by participants).

A Cox proportional hazards model was used to explore the factors associated with return to work time, and the assumptions of the model were tested. Baseline and operative variables were assessed in univariable analyses and those which were significant (p<0.05) were included as covariates in the final model. All regression analyses were adjusted for age and sex.

Participants were defined as having a poor outcome if they reported one or more of the following: global rating of change score of 'worse', 'unchanged' or 'slightly improved' (12 weeks after CTR)[20]; scar symptoms described as 'unbearable', 'very troublesome' or 'fairly troublesome' (12 weeks after CTR); use of antibiotics for an incision site infection after returning to work and additional sick-leave related to the CTR after returning to work. The duration of work absence for those with/without poor outcomes were compared using Wilcoxon rank-sum test. In addition, participants were dichotomised to those who returned to work within/after seven,

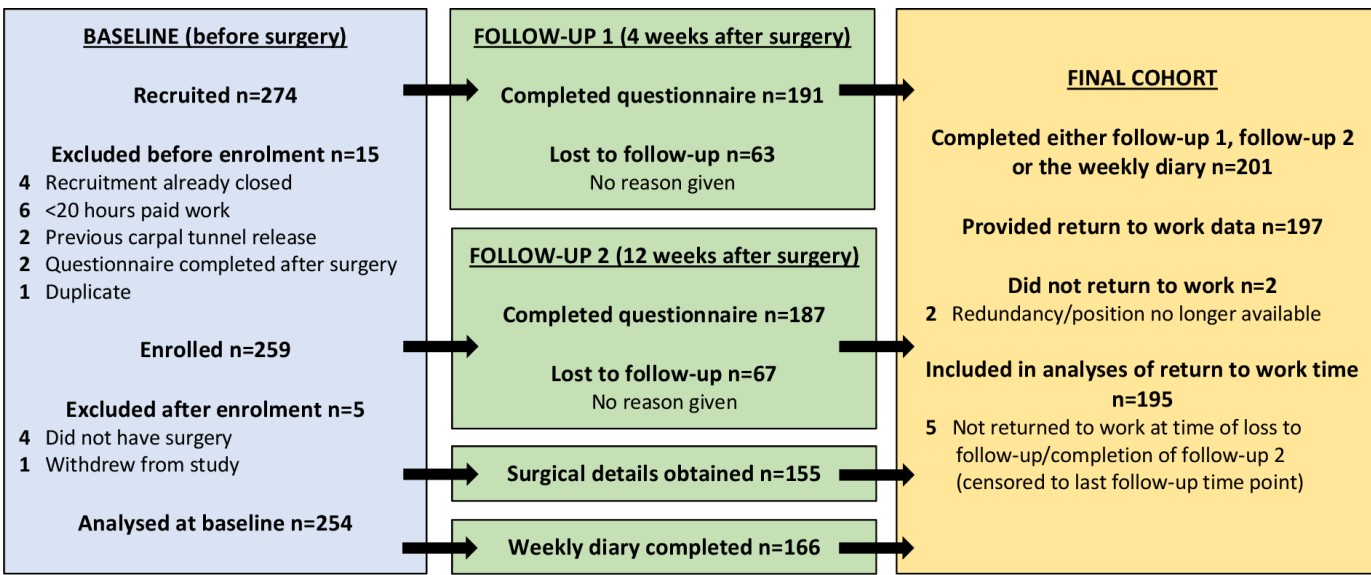

**Figure 1** Participant numbers for each stage of the cohort study.

14 and 28 days of surgery and the prevalence rates of poor outcomes were compared using $\chi^2$ for each time period. These time points were based on the median clinician-recommended return to work time that we reported previously.[6]

There was no imputation for missing data. Missing values were coded as a separate category for each of the variables included, and participant numbers are provided for each variable in the accompanying tables.

## RESULTS

A total of 254 individuals completed the baseline questionnaire and 201 (79%) provided follow-up data. Participant numbers and loss to follow-up for each study component are shown in figure 1. Participant demographics and comparisons between those who remained in the study and those who dropped out before follow-up are shown in table 1.

Eighty-six participants (43%) were recruited preoperatively on the day of their CTR. For the remaining 115 participants, the median time between recruitment and CTR was 14 days (IQR 5–40). The first follow-up questionnaire was completed a median of 32 days after CTR (IQR 29–38) and the final questionnaire was completed a median of 92 days after CTR (IQR 86–105). All participants underwent open CTR as a day case, and all but two had unilateral surgery. Another two participants required median nerve neurolysis, and one participant was noted to have a vascular abnormality. Sixty-two per cent of participants (n=125) were expecting to have a CTR for their other hand in the future. Other baseline demographic and surgical factors are shown in table 2.

The majority of participants (62%) worked 5 days per week (range 2–7) and the median number of weekly paid work hours was 37.5 (IQR 31–45). Two participants (1%) did not return to work during the 12-week study period: one reported that they had been made redundant and

the other that their job was no longer available. Four participants (2%) had incomplete data (missing return to work date or CTR date) meaning that the duration of work absence could not be calculated. These six individuals were not included in the analyses of return to work time, leaving a total sample size of 195. An additional five participants reported that they had not returned to work, but planned to do so in the future. These individuals were included in the return to work analysis, censored to the time of last follow-up.

The median duration of work absence after CTR was 20 days (IQR 12–33). Manual workers took longer to return than non-manual workers: 23.5 days (IQR 14–41) compared with 18 days (IQR 9–31). Those who were self-employed returned to work earlier than those who were employed: 13 days (IQR 6–19) compared with 22 days (IQR 14–38). Return to work times are shown in figure 2. The majority of participants returned to work on a Monday (43%). Approximately 15% returned each day between Tuesday-Thursday, then ~5% returned each day from Friday to Sunday. More than half of participants (59%) reported that they needed to modify their work duties to some extent when they first returned to work. Of these, 62% had resumed full duties within 5 weeks.

Univariable analyses of the relationship between baseline factors and the duration of work absence found 17 factors (age adjusted and sex adjusted) that were significantly associated with time to return to work and were entered into the multivariable model, in which five factors remained significant (table 3). Sensitivity analyses confirmed that these factors remained independently significant in the model. Non-significant findings in the univariable analyses are provided in online supplemental material 4. Having surgery in primary care and having a job with more than 4 hours of daily computer use were both associated with earlier return to work than their respective reference categories. Being female and having

**Table 2** Participant demographic and surgical factors

| | No of participants n=201 (%) |
|---|---|
| **Age (years)** | |
| 26–40 | 23 (11.4) |
| 41–55 | 101 (50.3) |
| ≥55 | 77 (38.3) |
| **Hand dominance** | |
| Right | 178 (88.6) |
| Left | 18 (9.0) |
| Ambidextrous | 5 (2.5) |
| **Side of surgery*** | |
| Dominant hand | 134 (66.7) |
| Non-dominant hand | 65 (32.3) |
| Bilateral surgery | 2 (1.0) |
| **Surgical specialty and grade** | |
| Consultant (plastics/orthopaedics) | 64 (31.8) |
| Registrar (plastics/orthopaedics) | 33 (16.4) |
| General practitioner | 62 (30.9) |
| Not reported | 42 (20.9) |
| **Incision type†** | |
| Mini open incision | 129 (64.2) |
| Traditional incision | 2 (1.0) |
| Not reported | 70 (34.8) |
| **Suture material** | |
| Absorbable | 24 (11.9) |
| Non-absorbable | 126 (62.7) |
| Not reported | 51 (24.4) |

*Considered as the non-dominant hand for those who reported ambidexterity.
†Mini open incision defined as distal to the distal wrist crease and traditional open excision extending proximally.

access to more than a month of paid sick leave were both associated with longer durations of work absence than their respected reference categories. Compared with those who expected to return within a week, there was a sequential increase in the likelihood of longer durations of work absence for those expecting to return between 7–14 days, 15–30 days and >30 days (table 3), which showed a significant gradient effect (p<0.001). The assessment of $R^2$ indicated that 46% of variation in the duration of work absence was explained by the model ($R^2$=0.46, 95% CI 0.37 to 0.53).

Clinical outcomes after CTR are shown in table 4. Using the definition outlined in the methods, a total of 46 participants (24%) were identified as having at least one poor outcome (CTS symptoms that were worse, unchanged or only slightly better; scar symptoms that were at least fairly troublesome; required postoperative antibiotics or

had additional time off work after first return). Of these participants, the majority (n=38, 83%) reported only a single component of poor outcome. Three individuals defined as having a poor outcome had not returned to work at the point of last follow-up (as compared with two individuals in the rest of the study sample). For those who had returned to work, the median duration of work absence for those with a poor outcome was 22 days (IQR 12–42) compared with 19 days (IQR 12–32) for those without (figure 2). This difference was not significant (Wilcoxon rank-sum test, p=0.24).

There was no significant difference in the prevalence of a poor outcome among those who returned to work within or after 7 days of CTR (20% vs 24%, $X^2$ p=0.63). Similarly, there was no significant difference in the prevalence of a poor outcome among those who returned to work within or after 14 days (19% vs 25%, $\chi^2$ p=0.31), or within or after 28 days of CTR (21% vs 27%, $\chi^2$ p=0.33).

## DISCUSSION

In this multicentre prospective cohort study, the median duration of work absence was 20 days (range 1–99), a duration similar to that reported by other European studies.[21] Earlier return to work was associated with typing for ≥4 hours at work (as compared with more physical workplace tasks) and undergoing surgery in primary care (as compared with secondary care or private practice). Preoperative expectations about return to work were important significant predictors of actual return to work times. We found no evidence of poorer clinical outcomes in the first 12 weeks among those who returned to work earlier. At each time point, fewer manual workers had returned to work than non-manual workers and fewer employed workers had returned than self-employed (figure 2). Both findings have been reported previously,[10 11 22 23] however, neither the type of work (manual/non-manual) nor the type of work contract (employed/self-employed) were significantly associated with the duration of work absence in the mutually adjusted model. These results illustrate the importance of considering the range of relevant demographic, clinical and occupational factors, which may have been confounders, moderators or mediators in previous studies. The reported model has not been developed to predict the duration of work absence for future CTR patients, rather to explore and identify important risk factors for consideration in future research.

Five variables remained statistically significantly associated with longer duration of work absence in the final model. Two were occupational factors: infrequent computer use and availability of sick pay. Cowan et al[10], recorded earlier return to work after CTR for desk-based workers and we have shown previously that UK hand surgeons and hand therapists report that they advise earlier return to desk-based workers.[6] The relationship between longer duration of work absence and availability of sick pay has also be reported previously for those with and without worker's compensation.[9 21] It is plausible

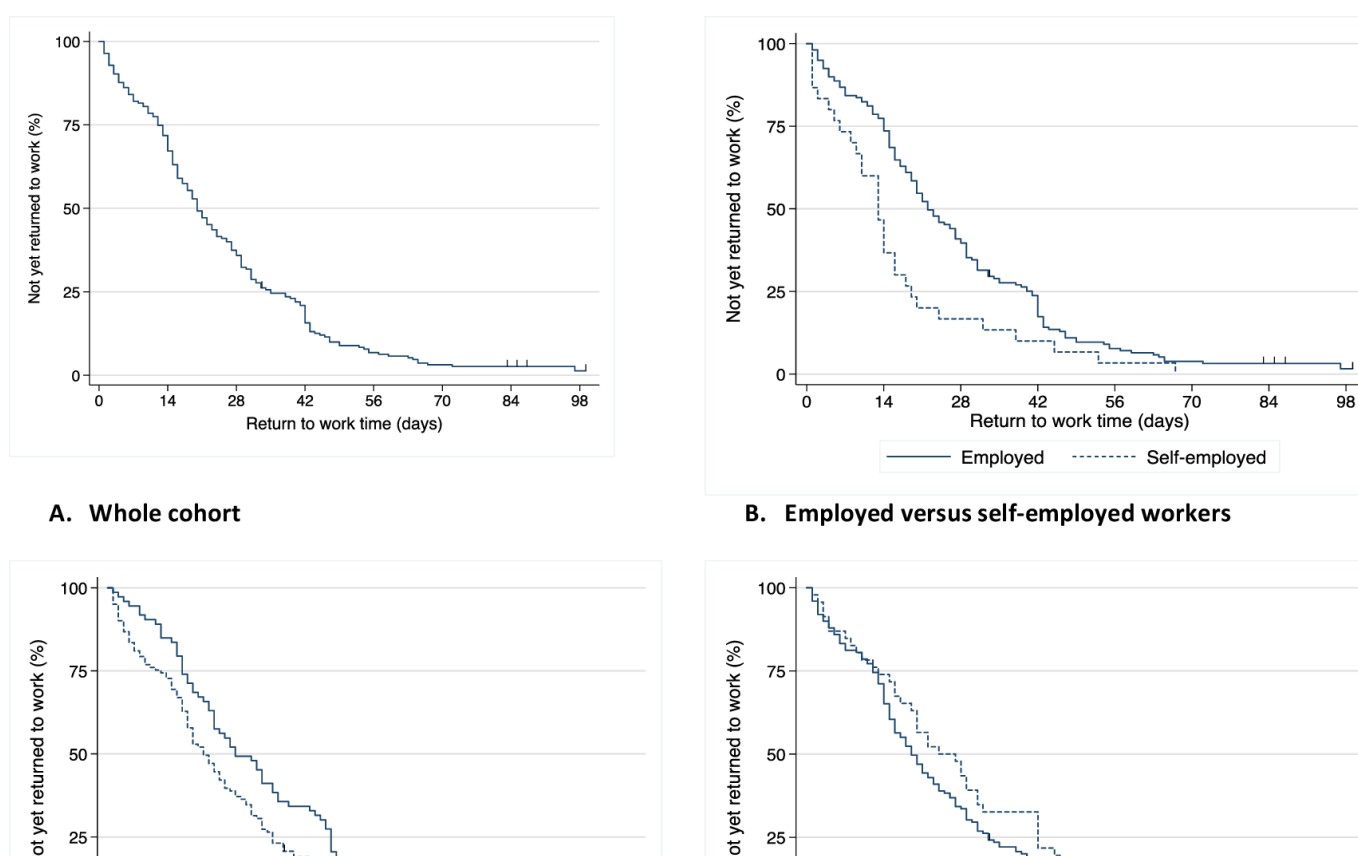

**A. Whole cohort**

**B. Employed versus self-employed workers**

**C. Non-manual versus manual workers**

**D. With versus without a poor outcome**

Vertical lines indicate right censoring for those who had not returned to work at the point of last follow-up

**Figure 2** Duration of work absence after carpal tunnel release.

that financial necessity is driving earlier return to work for those with limited sick pay. Alternatively, those with access to paid leave might choose to take the maximum available duration.

Only one clinical factor was significantly associated with the duration of work absence: participants who had their CTR surgery in primary care were more likely to return to work earlier than those whose procedure took place in an NHS hospital setting. One possible reason is that patients with more complex disease may be more commonly referred to secondary, rather than primary, care for their CTR and these patients may require longer off work after their surgery. However, in the current study, the proportion and degree of comorbidities, and the extent of preoperative symptoms reported by participants were similar across all settings. Another possibility is that the patients' expectations of the surgery may be different: CTR performed in a primary care setting might be perceived by patients as being more minor than surgery in a hospital operating theatre. Alternatively, the general practice surgeons may have recommended

earlier return to work than those based in a hospital, although the median expected duration of work absence for participants in the current study was the same across all settings. The inclusion of CTR performed in primary care is a strength of the study. We acknowledge that hand surgeons may also provide CTR services in primary care, as visiting clinicians, however, in the current study, this was not the case. CTR and other surgical procedures, such as vasectomy and minor skin surgery, are regularly performed by trained general practitioners in the UK,[24] and all primary care surgeons in the current study were general practitioners who already provided a CTR service.

Only one demographic factor was statistically significant: women were more likely to take longer to return to work than men. While we found inconsistent data about the effect of gender on return to work after CTR in the literature,[25 26] female gender has been associated with longer periods of work absence for other health conditions.[27] Researchers should continue to include sex as a covariate in analyses of return to work outcomes, although there is currently insufficient evidence to support any difference

**Table 3** Cox proportional hazards analyses of the association between baseline demographic, clinical and occupational factors and the duration of work absence after carpal tunnel release

| | N | Median work absence (days) | IQR | Univariable analyses | | | Multivariable analysis | | |
| --- | --- | --- | --- | --- | --- | --- | --- | --- | --- |
| | | | | HR | 95% CI | P value | HR | 95% CI | P value |
| Sex (censored: 5 females; no missing data) | | | | | | | | | |
| Male | 72 | 17.5 | 8–31 | 1 | – | – | 1 | – | – |
| Female | 118 | 21.5 | 14–35 | 0.79 | 0.59 to 1.06 | 0.12 | 0.56 | 0.36 to 0.88 | **0.01** |
| Age in years (censored: 1 aged 26–40, 3 aged 41–55, 1 aged >55; no missing data) | | | | | | | | | |
| 26–40 | 21 | 20 | 15–30 | 1.01 | 0.63 to 1.63 | 0.96 | 1.44 | 0.82 to 2.55 | 0.21 |
| 41–55 | 94 | 20 | 9–33 | 1 | – | – | 1 | – | – |
| >55 | 75 | 17 | 12–35 | 1.03 | 0.76 to 1.40 | 0.83 | 1.15 | 0.80 to 1.65 | 0.44 |
| Smoking status (censored: 1 never, 4 current/ex; 2 missing) | | | | | | | | | |
| Never | 105 | 16 | 12–31 | 1 | – | – | 1 | – | – |
| Current/ex | 83 | 23 | 13–41 | 0.74 | 0.56 to 1.00 | 0.046 | 0.75 | 0.51 to 1.09 | 0.13 |
| Site (censored: 5 NHS secondary care; no missing data) | | | | | | | | | |
| NHS primary care | 72 | 19.5 | 13–33 | 1.18 | 0.87 to 1.62 | 0.29 | 1.54 | 1.05 to 2.25 | **0.03** |
| NHS secondary care | 92 | 20 | 12–39.5 | 1 | – | – | 1 | – | – |
| Private facilities | 26 | 20 | 7–28 | 1.63 | 1.04 to 2.54 | 0.03 | 1.46 | 0.87 to 2.44 | 0.15 |
| Afraid of long-term hand problems* (censored: 1 no, 4 yes; 3 missing) | | | | | | | | | |
| No | 105 | 19 | 12–31 | 1 | – | – | 1 | – | – |
| Yes | 82 | 20.5 | 13–42 | 0.69 | 0.51 to 0.93 | 0.01 | 0.93 | 0.67 to 1.30 | 0.68 |
| CTS-6 score for side of surgery (tertiles)† (censored: 2 intermediate, 3 poor; 8 missing) | | | | | | | | | |
| Good (1–3.0) | 65 | 16 | 10–27 | 1 | – | – | 1 | – | – |
| Intermediate (3.2–3.8) | 58 | 21.5 | 14–35 | 0.77 | 0.54 to 1.10 | 0.15 | 1.19 | 0.77 to 1.84 | 0.44 |
| Poor (3.8–5) | 59 | 24 | 13–41 | 0.67 | 0.47 to 0.97 | 0.03 | 1.04 | 0.65 to 1.66 | 0.87 |
| Type of work contract (censored: 5 employed, 1 missing) | | | | | | | | | |
| Employed (permanent) | 154 | 22 | 14–38 | 1 | – | – | 1 | – | – |
| Self-employed | 30 | 13 | 6–19 | 1.72 | 1.13 to 2.61 | 0.01 | 1.19 | 0.67 to 2.14 | 0.55 |
| Zero hours/temporary | 5 | 12 | 3–31 | 2.01 | 0.81 to 5.00 | 0.13 | 0.73 | 0.25 to 2.14 | 0.56 |
| Duration of available sick pay (censored: 4 >1 month, 1 unsure; no missing data) | | | | | | | | | |
| ≤1 month | 49 | 16 | 12–29 | 1 | – | – | 1 | – | – |
| >1 month | 88 | 27 | 15–42 | 0.59 | 0.41 to 0.85 | 0.004 | 0.46 | 0.28 to 0.76 | **0.002** |
| Unsure | 53 | 16 | 10–23 | 1.19 | 0.80 to 1.77 | 0.40 | 1.01 | 0.61 to 1.66 | 0.97 |
| Access to occupational health at work (censored: 1 no, 4 yes; 1 missing) | | | | | | | | | |
| No | 110 | 15.5 | 9–29 | 1.77 | 1.31 to 2.38 | <0.001 | 1.42 | 0.91 to 2.19 | 0.12 |
| Yes | 79 | 25 | 16–42 | 1 | – | – | 1 | – | – |
| Expected duration of leave after carpal tunnel release (days) (censored: 1 7–14, 2 15–29, 2 ≥30; no missing data) | | | | | | | | | |
| <7 | 35 | 4 | 2–12 | 1 | – | – | 1 | – | – |
| 7–14 | 75 | 16 | 13–26 | 0.23 | 0.15 to 0.36 | <0.001 | 0.27 | 0.16 to 0.45 | **<0.001** |
| 15–29 | 35 | 29 | 22–39 | 0.12 | 0.07 to 0.19 | <0.001 | 0.19 | 0.10 to 0.37 | **<0.001** |
| ≥30 | 45 | 42 | 21–44 | 0.08 | 0.05 to 0.14 | <0.001 | 0.12 | 0.06 to 0.23 | **<0.001** |
| MHQ work functioning score (tertiles)‡ (censored: 3 poor, 2 intermediate; no missing data) | | | | | | | | | |

Continued

**Table 3** Continued

| | N | Median work absence (days) | IQR | Univariable analyses | | | Multivariable analysis | | |
|---|---|---|---|---|---|---|---|---|---|
| | | | | HR | 95% CI | P value | HR | 95% CI | P value |
| Poor (0–55) | 67 | 20 | 13–35 | 0.68 | 0.47 to 0.98 | 0.04 | 0.83 | 0.50 to 1.40 | 0.49 |
| Intermediate (60-80) | 72 | 21 | 12.5–39.5 | 0.77 | 0.53 to 1.10 | 0.15 | 0.81 | 0.50 to 1.31 | 0.39 |
| Good (81-100) | 51 | 17 | 10–29 | 1 | – | – | 1 | – | – |
| Job satisfaction§ (censored: 5 satisfied; 1 missing) | | | | | | | | | |
| Satisfied | 169 | 19 | 11–31 | 1 | – | – | 1 | – | – |
| Dissatisfied | 19 | 38 | 21–43 | 0.61 | 0.38 to 0.99 | 0.04 | 0.67 | 0.38 to 1.16 | 0.15 |
| Believe that the hand problem was caused by work¶ (censored: 1 no, 4 yes; 2 missing) | | | | | | | | | |
| No | 112 | 19 | 13–31 | 1 | – | – | 1 | – | – |
| Agree/strongly agree | 76 | 23 | 10–42 | 0.62 | 0.46 to 0.85 | 0.003 | 0.82 | 0.57 to 1.17 | 0.28 |
| Job is demanding on hands/wrists* (censored: 5 yes; no missing data) | | | | | | | | | |
| No | 35 | 16 | 6–27 | 1 | – | – | 1 | – | – |
| Yes | 155 | 20 | 13–38 | 0.61 | 0.42 to 0.89 | 0.01 | 0.68 | 0.42 to 1.12 | 0.13 |
| Type of work** (censored: 2 non-manual, 3 manual; 1 missing) | | | | | | | | | |
| Non-manual | 119 | 18 | 9–31 | 1 | – | – | 1 | – | – |
| Manual | 70 | 23.5 | 14–41 | 0.66 | 0.48 to 0.89 | 0.01 | 0.97 | 0.57 to 1.64 | 0.90 |
| Work involves target-driven pay†† (censored: 3 no, 2 yes; 10 missing) | | | | | | | | | |
| No | 149 | 19 | 12–31 | 1 | – | | 1 | – | – |
| Yes | 31 | 22 | 13–45 | 0.61 | 0.41 to 0.91 | 0.02 | 0.97 | 0.59 to 1.61 | 0.91 |
| Duration of computer use at work (hours)†† (censored: 5 <1; four missing) | | | | | | | | | |
| <1 | 69 | 28 | 17–42 | 1 | – | – | 1 | – | – |
| >1 to <4 | 33 | 16 | 10–31 | 2.20 | 1.43 to 3.38 | <0.001 | 1.01 | 0.56 to 1.81 | 0.98 |
| ≥4 | 84 | 16 | 7–27 | 2.38 | 1.67 to 3.38 | <0.001 | 1.85 | 1.08 to 3.16 | **0.03** |
| Work involves lifting or carrying ≥10 kg (censored: 4 no, 1 yes; 5 missing) | | | | | | | | | |
| No | 108 | 18.5 | 11–30 | 1 | – | – | 1 | – | – |
| Yes | 77 | 24 | 13–40 | 0.61 | 0.42 to 0.86 | 0.01 | 0.80 | 0.48 to 1.33 | 0.39 |
| Work involves pushing/pulling a heavy weight†† (censored: 2 no, 3 yes; 2 missing) | | | | | | | | | |
| No | 104 | 16 | 8.5–28.5 | 1 | – | – | 1 | – | – |
| Yes | 83 | 26 | 16–42 | 0.51 | 0.37 to 0.70 | <0.001 | 0.97 | 0.61 to 1.55 | 0.90 |

Total sample size n=195. Median work absence relates to the 190 non-censored events. All analyses were adjusted for age and sex. All significant variables in the univariable analyses (p<0.05) were entered into the multivariable model. Significant variables in the multivariable analysis are indicated in bold.

*Reported on a 0–10 scale, dichotomised to no (0–6) and yes (7–10).[13]
†CTS-6 score[14] with data-driven tertiles.
‡MHQ work performance subscale scored from 0 to 100, higher scores indicate better perceived work performance.[17] Data driven tertiles.
§Reported on a 5-point scale, dichotomised to satisfied (very satisfied/satisfied/fairly satisfied) and dissatisfied (dissatisfied/very dissatisfied).
¶Reported on a 5-point scale and dichotomised to agree/strongly agree and neither agree nor disagree/disagree/disagree strongly.[36]
**Classified using the Office for National Statistics Standard Occupational Classification 2010.[18 19]
††As part of the normal working day.[33]
CTS, carpal tunnel syndrome; MHQ, Michigan Hand Questionnaire ; NHS, National Health Service.

in return to work recommendations after CTR based on sex. Further qualitative investigation is required in order to understand the context for this.

Finally, those who expected to return to work more quickly were significantly more likely to do so. It has been shown previously that patient expectations are a

**Table 4** Clinical outcomes after carpal tunnel release

| | Mean score (SD) | |
| --- | --- | --- |
| | Before surgery | 12 weeks after surgery |
| CTS-6 (operated hand) * | 3.3 (0.87) | 1.2 (0.54) |
| MHQ function (operated hand) † | 50 (22.1) | 79 (19.4) |
| MHQ satisfaction with function (operated hand) † | 38 (25.7) | 82 (21.3) |
| MHQ bilateral activities of daily living † | 69 (23.7) | 88 (13.8) |
| MHQ activities of daily living (operated hand) † | 65 (28.1) | 87 (18.5) |
| | No of participants (%) | |
| Global rating of change score | | |
| Completely cured | – | 65 (33.3) |
| Much better | – | 98 (50.3) |
| Slightly better | – | 13 (6.7) |
| Unchanged | – | 2 (1.0) |
| Worse | – | 5 (2.6) |
| Scar symptom severity | – | |
| Not at all troublesome | – | 99 (50.8) |
| A little troublesome | – | 63 (32.3) |
| Fairly troublesome | – | 18 (9.2) |
| Very troublesome | – | 2 (1.0) |
| Unbearable | – | 0 |
| Required postoperative antibiotics | – | 10 (5.1) |
| Additional sick leave after first returning to work | – | 12 (6.2) |

Grey shading indicates the categories, which were combined to define a poor surgical outcome.
*CTS-6 assessment of carpal tunnel syndrome symptoms.[14] Range 1–5: 1 equals no symptoms.
†MHQ Michigan Hand Questionnaire.[17] Range 0–100: 100 equals no deficit or dissatisfaction.
CTS, carpal tunnel syndrome; MHQ, Michigan Hand Questionnaire.

prominent determinant of return to work time, or other return to work outcomes, for musculoskeletal or mental health conditions.[27–29] The role of expectations on outcomes, including the expected and actual timing for return to work and driving after hand and wrist surgery, requires further exploration, particularly because expectations are a potentially modifiable characteristic which could be influenced by the advice provided by clinicians preoperatively.

In total, approximately a quarter of participants in this study were considered to have a poor outcome using our composite definition. We chose to use a multicomponent definition, which was deliberately very stringent, to minimise the chances of missing any harm caused by early return to work. Our rates of poorer outcomes were in fact

similar to those reported in other CTR populations.[20 30 31] Importantly, we found no relationship between earlier return to work and occurrence of poor outcomes within 12 weeks of CTR in this cohort study. We acknowledge that a longer follow-up duration would have aided the assessment of postoperative symptom resolution, however, this was not possible with the resources available and was not a primary objective of the study.

There are a number of limitations of the current study, including the reliance on self-reported data. Work absence is not logged on a national database in the UK and therefore could only be obtained through self-report. To minimise errors of recall, date of return to work was determined contemporaneously. The recall duration for measures of function and symptoms was limited to a maximum of 4 weeks, consistent with the outcome measures used.[14 17] We set out to recruit a large sample of working-aged adults undergoing CTR. Our prospectively recruited sample from 16 sites is one of the largest reported in the literature to date, with a good follow-up response rate (79%), but it remains possible that we were underpowered to detect some of the factors which may have been associated with delayed return to work. Specifically, this could result where some levels of categorical variables of interest have lower prevalence, for example, the type of work contract (>80% of participants reported that they had a permanent work contract, compared with ~15% who were self-employed). Furthermore, we acknowledge that the inclusion of a large number of variables in the development of the final model may result in model overfitting, thereby potentially limiting generalisability.

We took the approach not to impute values where data were missing. Overall, the amount of missing data was small and at the individual item level (table 3 and online supplementary material 4). Missing data were coded as such, and included in the analysis. We acknowledge that the approach taken to missing data may have resulted in biased estimates, yet if such effects are present, they are likely to be minimal due to low levels of missing data.

Following our a priori analysis plan, the association between each baseline variable and the duration of work absence was individually assessed in separate age-adjusted and sex-adjusted analyses. Only those variables which reached significance at the 5% level ($p < 0.05$) were included in the multivariable model. In order to test the stability of our model, and to identify whether any potential associations had been missed, this was tested using 1% and 20% cut-offs. In both test scenarios, the findings were similar to those presented in our final model (table 3), suggesting that our model is robust. However, we acknowledge that alternative methods of selecting variables for inclusion (such as forward inclusion or backward elimination) may have yielded slightly different results, particularly for variables that were close to our significance cut-off of 5%.

The findings may not be generalisable to working populations in regions outside of central and southern England and Wales, who are employed in other industries,

or managed with a different CTR patient pathway. Steps were taken to include the main settings where CTR is performed in the UK, but we acknowledge that CTR may also be performed by other specialties. Individuals who chose to participate in the study may not be fully representative of the wider CTR population, and the observed differences between those who completed the study and those who were lost to follow-up (younger, poorer mental health, more likely manual workers) also limit generalisability. Furthermore, we acknowledge that our model explained only 46% of variation in the duration of work absence.

Endoscopic CTR has been associated with earlier return to work than open CTR,[32] however, it was not possible to assess this in the current study. At present, endoscopic CTR is not routinely performed in the UK.[6] Anecdotally, most providers will not fund the extra cost of endoscopic CTR, which requires extra equipment, longer operating times and more experienced surgeons. Recruitment to the current study was not limited to patients undergoing open CTR, but no endoscopic procedures were performed during the study at any of our sites.

All participants were presumed to have CTS as diagnosed by their treating clinician. Many studies of CTS include nerve conduction study (NCS) findings as part of their eligibility criteria, although this was not possible in the current study because NCS are not routinely recommended for pre-operative diagnosis of CTS in the UK.[2] Our eligibility criteria required that only people undergoing their first CTR were included and reported on in this study (so that previous experiences with CTR were not potential confounders). However, more than three-quarters of the cohort reported bilateral symptoms. The possible impact of persisting CTS symptoms in the non-operated hand on return to work also needs to be considered.

For the current study, we considered both occupational title and self-reported occupational exposures collected in a standardised questionnaire format.[13 18 19 33] Categorisation based on job title and industry may not accurately reflect the physical and/or psychosocial aspects of job role. Furthermore, co-occurrence of occupational exposures may be more common in some types of jobs than in others, for example, lifting >10 kg and pushing or pulling a heavy weight.

There is a need for an agreed approach to identifying and recording key physical demands and psychosocial exposures of jobs to enable consistent exploration of their impact on work and clinical outcomes following surgery or other intervention. Approaches such as job exposure matrices[34] could facilitate this in future research.

In summary, this large multicentre prospective cohort study investigated when participants return to work after CTR. Expectations about return to work (reported before surgery) were strongly associated with actual work absence, regardless of the job role or self-reported upper limb activities involved. Patient expectations can be influenced by many factors, but one of the most important is the advice provided by clinicians, in particular the surgeon. This suggests that clear, consistent advice could have an important effect on duration of sick leave. To date, there is no evidence-based guidance informing clinicians what to advise about returning to different types of work after CTR. Further research is required to reach a consensus and explore whether the provision of targeted, consistent and standardised advice can alter the expected duration of work absence, reducing unnecessary sick leave, without causing adverse effects on clinical outcomes.

**Acknowledgements** The authors would like to thank and acknowledge all the study participants who gave up their time to take part in this study. Thank you also to the Patient Advisory Group for their contribution to the development of the research question and study material, and their thoughtful feedback on the preliminary findings. Thank you to Professor David Coggon for his assistance with the initial research idea, and to the Association of Surgeons in Primary Care and British Society for Surgery of the Hand for supporting the study. Thank you to the study sites and local principal investigators: Jeremy Bland at East Kent Hospitals University NHS Foundation Trust; Darren Roberts at Portsmouth Hospitals NHS Trust; Mike Taylor and Claire Zweifel at Mid Essex Hospital Services NHS Trust; Frances Clark at Tollgate Clinic; Will Mason at Gloucester Hospitals NHS Foundation Trust; Vasileios Kefalas and Petros Mikalef at Care UK Southampton NHS Treatment Centre; Peter Sharpe at Salisbury Medical Practice; David Warwick at University Hospital Southampton NHS Foundation Trust and Nuffield Health Wessex Hospital; Alistair Phillips at Southern Health NHS Foundation Trust; Nanda Pillai at St Luke's Surgery, Walsall; Tim Halsey and Effie Katsarma at Chelsea and Westminster Hospital NHS Foundation Trust; Claire Middleton at Royal Berkshire NHS Foundation Trust; Duncan Avis and Ian MacLeod at Hampshire Hospital NHS Foundation Trust; Nick Gape and Lisa Small at Cardiff and Vale University Health Board; Kristin Francis at HCA Healthcare London Hand and Wrist Unit; and to all local team members who helped set up and recruit to the REACTS study.

**Contributors** The study was devised by LN, KW-B, JA and DW. LN completed data collection and analysis, with assistance from GN and KW-B. All authors contributed to the interpretation of the data. LN wrote the first draft of the manuscript. All authors reviewed and edited the manuscript and approved the final version.

**Funding** LN was supported by a National Institute for Health Research (NIHR) Doctoral Research Fellowship, grant number DRF-2015-08-056. GN was supported by funding from the MRC Versus Arthritis Centre for Musculoskeletal Health and Work, grant reference 22090.

**Conflicts of interest** LN was supported by a National Institute for Health Research (NIHR) Doctoral Research Fellowship. GN was supported by funding from the MRC Versus Arthritis Centre for Musculoskeletal Health and Work. The authors declare no other conflicts of interest.

**Map disclaimer** The views expressed are those of BMJ Open, the authors, and not necessarily those of the funders, the NHS, the NIHR or the Department of Health.

**Competing interests** None declared.

**Patient consent for publication** Written informed consent was obtained from the patient(s) for their anonymised information to be published in this article.

**Ethics approval** Full ethics approval was granted by the NHS Health Research Authority (IRAS 209840: 16/WA/0390) and University of Southampton (ERGO 25757) Ethics Committees.

**Provenance and peer review** Not commissioned; externally peer reviewed.

**Data availability statement** Data are available on reasonable request. The anonymised dataset is held at the University of Southampton: https://eprints.soton.ac.uk/436526/.

**ORCID iDs**
Lisa Newington http://orcid.org/0000-0001-6954-2981
Georgia Ntani http://orcid.org/0000-0001-7481-6860
David Warwick http://orcid.org/0000-0003-3030-442X
Jo Adams http://orcid.org/0000-0003-1765-7060
Karen Walker-Bone http://orcid.org/0000-0002-5992-1459

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
