## [Reviewer comments · BMJ Open]

ARTICLE DETAILS

TITLE (PROVISIONAL)	Sickness absence after carpal tunnel release: a multi-centre prospective cohort study
AUTHORS	Newington, Lisa; Ntani, G; Warwick, David; Adams, Jo; Walker-bone, Karen

VERSION 1 – REVIEW

REVIEWER	Dr Claire Burton Keele University, UK
REVIEW RETURNED	03-Sep-2020

GENERAL COMMENTS	This study asks an important question for patients and the clinicians treating them, providing evidence to support post-operative care and facilitate a timely return to the workplace. Clarifications and changes that would further improve this manuscript mainly concern the development and reporting of the prognostic model, specifically: 1. to describe the processes involved in identifying candidate predictors and the handling of missing data2. to consider whether univariable screening was the most appropriate way to select candidate predictors for inclusion in the multivariable model. It would be helpful to present the final model and report its performance and describe any attempt to validate the model.3. to discuss the appropriateness of the sample size given the number of candidate predictors considered4. to consider whether significance testing in Table 2 is required5. to describe the methods, if any, were used to follow up non-respondents6. All participants underwent open CTR. Might the results have been different if an endoscopic approach had been used?7. Page 17 line 3 suggests that the five variables in the model were independently associated with outcome. The model will present the combined effect of the included variables.
---

REVIEWER	Jane E McEachan Queen Margaret Hospital, UK
REVIEW RETURNED	27-Sep-2020

GENERAL COMMENTS	This is a good piece of work. The paper is interesting and of significant importance when counselling patients on the time off work required. The use of 3 month outcomes raises issues regarding whether or not this predicts outcome at 1 year- The overall improvement rate at 3 months seems low, and the factors driving
---

	this are unclear - In my experience, a number of patients with ongoing discomfort from loss of the flexor retinaculum often report negatively, when at 1 year the problem is no longer an issue, Overall, this is a well written, relevant and interesting paper.
--	---

REVIEWER	Descatha UNIV Angers, CHU Angers, Inserm, FRANCE
REVIEW RETURNED	13-Nov-2020

GENERAL COMMENTS	Very interesting study, and have some comments about methodology (I have asked to do this part of the review), that highlight my interest on the manuscript  1. I would advise to have more information of representativeness of the hand surgery centers in UK and more info on general organization for readers not in UK. The authors stated ‘Sites were NHS secondary care (hospital setting), NHS primary care and private hand surgery facilities, representing the range of UK healthcare facilities where CTR is performed’. Representativeness should be given, especially when you look the type of surgery where no endoscopic released have been reported, and 30% of GP perform de surgery. These might be very surprising for other countries. I would expect a comparison inside UK with NHS numbers in comparison, and the singularity of UK where non hand specialist might perform surgery? 2. In the same selection effects, what is convenient sample (needs to be described more), similarly to the loss to follow up 3. Though the authors stated “Sensitivity analyses confirmed that these factors were independently significant in the model with no collinearity”, I am still concerned of over adjustment (physical exposure for instance). Taking this example, when you pay attention to the HR, both significant of these two variables: Work involves pushing/pulling a heavy weight and Work involves lifting or carrying ≥10 kg, because they are probably very closely associated (and negatively associated with duration in computer use). 4. A related comment is the parsimony concept: with a number of subjects 201 and 19 variables in the cox model, I feel this is not the case. I would advise to select relevant variables before where proportional risk might also be confirm. Authors should stay on the variable that are supposed to be be associated and that make sense. NB In table 4, CTS6 = Poor (8.8-5) should be Poor (3.8-5)
--

VERSION 1 – AUTHOR RESPONSE

Reviewer: 1

Comments to the author

This study asks an important question for patients and the clinicians treating them, providing evidence to support post-operative care and facilitate a timely return to the workplace.

Clarifications and changes that would further improve this manuscript mainly concern the development and reporting of the prognostic model, specifically:

1. Describe the processes involved in identifying candidate predictors and the handling of missing data

Thank you for highlighting this point. Candidate predictors were predominantly identified from a previous systematic review (Peter's et al. 2016), which explored the prognostic factors for return to work following carpal tunnel release. We also developed the questionnaires in discussion with our PPI group, including their review of the importance of each question (variable) and potential item redundancy from a patient's perspective. We have added additional text in the methods section to ensure that this is clear for the reader and have included an additional file (now supplementary file 1) outlining the reasoning for each included variable.

Amended section: Methods (page 6)

Amended text: *Questionnaire content was informed by the clinical, demographic and occupational factors previously identified in a systematic review of prognostic factors for return to work after CTR[8] and developed in collaboration with our patient advisory group. The reasoning for item inclusion is provided in supplementary file 1.*

A statement outlining our analysis approach to missing data was initially included in the statistical methods section (i.e. "There was no imputation for missing data"). However, due to its location in the text, it may not have been apparent how this was applied, thank you for highlighting this lack of clarity. We have now changed the wording and location of this statement.

Amended section: Statistical methods (page 8)

Amended text: *There was no imputation for missing data. Missing values were coded as a separate category for each of the variables included, and participant numbers are provided for each variable in the accompanying tables.*

2. Consider whether univariable screening was the most appropriate way to select candidate predictors for inclusion in the multivariable model. It would be helpful to present the final model and report its performance and describe any attempt to validate the model.

Each variable was assessed in an age and sex adjusted model to explore whether there was an association with duration of work absence (presented in the supplementary tables). Those which were significant at the 5% level were included in the final model. We took this approach knowing that there would be too many variables to use a single mutually adjusted model. Our analysis plan was developed a priori. Alternative methods of analysis may have included forward selection or backward elimination of variables. A potential limitation of our model is that we may have missed associations by only including those variables which were significant at the 5% level.

In response to your comment, we reviewed the model using an initial significance level of 20% to assess whether additional associations were identified using this approach. Using this method, 31 variables were included in the multivariable model. The variables which remained significantly associated with the duration of work absence were similar to those identified in our original model: expected duration of work absence, hospital site, duration of available sick pay, computer use. In addition, this model found the following to be associated with work absence: type of work contract and sick leave (not related to CTS) in the last month (before completion of the baseline questionnaire). Both of these variables had very uneven distribution of participants within the respective categories (30 self-employed, 159 permanent contract, 5 zero hours/other contract; 19 had pre-operative sick leave, 156 didn't) and wide confidence intervals for the effect estimate.

The similarity of our findings using our model with a 5% significance cut off for inclusion as compared with the model described above suggests that our model is a robust representation of the data. We acknowledge that potential associations may have been missed and have added this point to the limitations of our study.

To validate the model during the initial analysis, we also tested it in a sensitivity analysis using an initial cut-off at a 1% significance level. Using this approach, 13 variables (including age and sex) progressed to the multivariable model, compared with 19 that were identified at the 5% level. The findings of this test version of the model were identical to that reported in our manuscript, with the exception of the finding that earlier return to work was associated with having the CTR performed in primary care, as this variable was not included in the test model.

In light of these findings, we are confident that our presented multivariable model is an appropriate and robust representation of the data. We have included additional information to support this, along with the limitations discussed above.

Amended section: Discussion (page 19-20)

Additional text: *Following our a priori analysis plan, the association between each baseline variable and the duration of work absence was individually assessed in separate age- and sex-adjusted analyses. Only those variables which reached significance at the 5% level ($p < 0.05$) were included in the multivariable model. In order to test the stability of our model, and to identify whether any potential associations had been missed, this was tested using 1% and 20% cut-offs. In both test scenarios, the findings were similar to those presented in our final model (Table 4), suggesting that our model is robust. However, we acknowledge that alternative methods of selecting variables for inclusion (such as forward inclusion or backward elimination) may have yielded slightly different results, particularly for variables that were close to our significance cut-off of 5%.*

3. Discuss the appropriateness of the sample size given the number of candidate predictors considered

We acknowledge that the sample size of ~200 means that there is the possibility that our study was underpowered and that potential associations may have therefore been hidden. With ~200 participants (all experiencing the event), the rule of thumb (of 10 events per variable) would suggest that up to ~20 variables would be acceptable in the model. It has also been suggested that 5-9 events per predictor may be enough (Vittinghoff, & McCulloch. Relaxing the Rule of Ten Events per Variable in Logistic and Cox Regression". American Journal of Epidemiology 2007;165(6):710–718).

In addition, the stability of our model demonstrated in response to comment 2 (above) suggests that the sample size is adequate for the number of variables included, but we do recognise (and have reported) the potential lack of power as a limitation.

Amended section: Discussion (page 19)

Amended text: *Our prospectively recruited sample from 16 sites is one of the largest reported in the literature to date, with a good follow up response rate (79%), but it remains possible that we were under-powered to detect some of the factors which may have been associated with delayed return to work. Specifically, this could result where some levels of categorical variables of interest have lower prevalence, for example, the type of work contract (>80% of participants reported that they had a permanent work contract, compared with ~15% who were self-employed).*

4. Consider whether significance testing in Table 2 is required

Thank you for raising this point. Significance testing was included to aid our assessment of statistically, in addition to clinically, significant differences between groups, particularly for the continuous variables. We agree that it is not common practice to test for significance in analyses exploring potential sources of bias. We have removed the significance testing column from Table 2 and the text relating to significance testing in the statistical methods.

5. Describe the methods, if any, were used to follow up non-respondents

Thank you for identifying that this information was missing from our manuscript.

Amended section: Methods (page 7)

Additional text: *Steps were taken to minimise loss to follow-up after recruitment. To maximise retention, we incentivised with a shopping voucher (£10) on completion of the study and we sent up to three reminders using a combination of post, email and text.*

6. All participants underwent open CTR. Might the results have been different if an endoscopic approach had been used?

This is an interesting question, and the published literature does suggest that endoscopic CTR is associated with earlier return to work when compared with open CTR (most recently: Li et al *BMC Musculoskel Disord* 2020;21:270 doi <https://doi.org/10.1186/s12891-020-03306-1>). However, our survey of UK practice found that the majority of the 173 consultant hand surgeon respondents currently perform open CTR (Newington et al. *J Hand Surg Eur* 2018;43(8):875-78 doi:10.1177/1753193418786375). We specifically asked about endoscopic CTR, and 97% of respondents stated that they did not perform this procedure. We did not specifically exclude endoscopic CTR from the current study, but no participants were treated with this surgical approach. Anecdotally, most providers in the UK (NHS and insurance) will not fund the extra cost of endoscopic carpal tunnel release which requires extra equipment, longer operating time and more experienced surgeons. We have highlighted this in the discussion.

Interestingly, in the current study we found that the expected duration of work absence (reported pre-operatively) was strongly associated with the duration of work absence. It could be that a similar process occurs with endoscopic CTR, such that patients are advised to return to work more quickly than those undergoing open CTR, and accordingly, they do so. Unfortunately, this is not explored in the existing literature.

Amended section: Discussion (page 20)

Additional text: *Endoscopic CTR has been associated with earlier return to work than open CTR [35], however it was not possible to assess this in the current study. At present, endoscopic CTR is not routinely performed in the UK[6]. Anecdotally, most providers will not fund the extra cost of endoscopic CTR, which requires extra equipment, longer operating times and more experienced surgeons. Recruitment to the current study was not limited to patients undergoing open CTR, but no endoscopic procedures were performed during the study at any of our sites.*

7. Page 17 line 3 suggests that the five variables in the model were independently associated with outcome. The model will present the combined effect of the included variables.

Thank you for this comment and we apologise that the wording in this sentence appeared misleading. As all variables are included in a single mutually adjusted model, the effect estimate for any given exposure is interpreted as the risk of the outcome if exposed to the given exposure keeping all other variables constant, or in other words, the risk of the outcome if exposed to the given exposure is independent of the effect of the other exposures in the model. However, to avoid confusion, we have made the following changes:

Amended section: Discussion (page 17)

Amended text: *Five variables remained statistically significantly associated with longer duration of work absence in the final model.*

In addition, the statement inferring that our sensitivity suggested no collinearity has been removed.

Amended section: Results (page 12)

Amended text: *Sensitivity analyses confirmed that these factors remained significant in the model*

Reviewer: 2

Comments to the Author

This is a good piece of work. The paper is interesting and of significant importance when counselling patients on the time off work required.

1. The use of 3 month outcomes raises issues regarding whether or not this predicts outcome at 1 year- The overall improvement rate at 3 months seems low, and the factors driving this are unclear - In my experience, a number of patients with ongoing discomfort from loss of the flexor retinaculum often report negatively, when at 1 year the problem is no longer an issue. Overall, this is a well written, relevant and interesting paper.

Thank you for your comments. We agree that an extra follow-up time point would have added additional strength to the study. Unfortunately, due to limitations of both time and money, this was not possible. We have highlighted this as a limitation. We agree that in our clinical experience, it would be rare to expect persisting post-operative issues 12 months after CTR.

Three months was chosen as the final follow-up time point because we anticipated that the majority of patients would have returned to work by this time, and this was our primary outcome. Only 5 participants had not returned to work by 3 months, which, we feel, justifies our choice of timing.

Amended section: Discussion (page 19)

Additional text: *We acknowledge that a longer follow-up duration would have aided the assessment of post-operative symptom resolution, however this was not possible with the resources available and was not a primary objective of the study.*

Reviewer: 3

Comments to the Author

Very interesting study, and have some comments about methodology (I have asked to do this part of the review), that highlight my interest on the manuscript

1. I would advise to have more information of representativeness of the hand surgery centers in UK and more info on general organization for readers not in UK. The authors stated ‘Sites were NHS secondary care (hospital setting), NHS primary care and private hand surgery facilities, representing the range of UK healthcare facilities where CTR is performed’. Representativeness should be given, especially when you look the type of surgery where no endoscopic released have been reported, and 30% of GP perform de surgery. These might be very surprising for other countries. I would expect a comparison inside UK with NHS numbers in comparison, and the singularity of UK where non hand specialist might perform surgery?

Thank you for highlighting the need to clarify this for international readers. Provision of CTR is subject to many influences of which we are aware, but cannot quantify as data are not collected within the health system. As discussed in response to reviewer 1 (point 6), endoscopic CTR is not routinely performed in the UK. To the best of our knowledge, we believe there is currently only one NHS hand unit where endoscopic CTR is offered by one or two surgeons. Endoscopic CTR was not excluded, rather only open CTR was performed at the 16 sites. We have added this to the limitations of the study (please see response to reviewer 1 above).

The inclusion of NHS secondary care, primary care and private settings was chosen to represent the spread of CTR experiences for patients in the UK. This represents our experience (and networks) as practising clinicians in different areas of the country. Unfortunately, there are no available data supporting the numbers of CTRs performed in each setting because this crosses different healthcare systems. Preliminary work for this study led us to explore the different settings with support from the British Society for Surgery of the Hand and Association for Surgeons in Primary Care. This has been added to the manuscript in the methods. We appreciate that CTR may occur in other surgical settings (for example general orthopaedics), but these were not possible to access. This has been added to the limitations.

Amended section: Methods (page 5-6)

Additional text: *Provision of CTR in the UK was explored through discussion with relevant national organisations (British Society for Surgery of the Hand and Association for Surgeons in Primary Care). Sites were recruited through National Institute for Health Research infrastructure.*

Amended section: Discussion (page 17)

Additional text: *The inclusion of CTR performed in primary care is a strength of the study. CTR and other surgical procedures, such as vasectomy and minor skin surgery, are regularly performed by trained General Practitioners in the UK[27].*

Amended section: Discussion (page 20)

Amended text: *Steps were taken to include the main settings where CTR is performed in the UK, but we acknowledge that CTR may also be performed by other specialities. Individuals who chose to participate in the study may not be fully representative of the wider CTR population, and the observed differences between those who completed the study and those who were lost to follow-up (younger, poorer mental health, more likely manual workers) also limit generalisability.*

2. In the same selection effects, what is convenient sample (needs to be described more), similarly to the loss to follow up

Thank you for this comment. The truly honest answer is that the convenience sample was as many patients as we could recruit given the time and resources available.

We acknowledge the important consideration of the small sample size of our study. However, the potential implication of the low statistical power is lack of ability to detect actual/real effects of exposures (Type II error) [Altman DG. Statistics and ethics in medical research III: how large a sample size? *Br Med J* 1980; 281: 1336-8 PubMed .]. In our study, we identified several exposures significantly associated with time return to work. The effect of these associations had the direction and the size one might expect. As such, even though we do appreciate the potential issues associated with small sample sizes, our findings indicate significant and plausible associations.

The discussion of potential selection and attrition bias has been extended in the limitations section, highlighting key differences between the sample and those lost to follow-up.

Amended section: Discussion (page 20)

Amended text: *Steps were taken to include the main settings where CTR is performed in the UK, but we acknowledge that this procedure may also be performed by other specialities. Individuals who chose to participate in the study may not be fully representative of the wider CTR population, and the observed differences between those who completed the study and those who were lost to follow-up (younger, poorer mental health, more likely manual workers) also limit generalisability.*

3. Though the authors stated “Sensitivity analyses confirmed that these factors were independently significant in the model with no collinearity”, I am still concerned of over adjustment (physical exposure for instance). Taking this example, when you pay attention to the HR, both significant of these two variables: Work involves pushing/pulling a heavy weight and Work involves lifting or carrying ≥ 10 kg, because they are probably very closely associated (and negatively associated with duration in computer use).

Thank you for these helpful comments. We have removed the phrase ‘no collinearity’, and accept that this was not appropriate.

Amended section: Results (page 12)

Amended text: *Sensitivity analyses confirmed that these factors remained significant in the model*

We appreciate the concern about over adjustment, particularly in relation to mutual adjustment of job activities. During the analysis, we ran cross-tabulations to see where intuitively co-occurring (or inversely occurring) exposures might overlap. This included the following occupational exposures:

22% of participants reported one, but not both of pushing/pulling a heavy weight and lifting or carrying >10kg.

10% of participants reported using a computer for >4 hrs and pushing or pulling a heavy weight, similarly and 10% reported using a computer for >4hrs and lifting >10kg.

To address this concern, we ran some further survival models keeping each of the three job activities one at a time, and separately with and without the type of job (manual/non-manual). In all cases,

neither pushing or pulling a heavy weight, nor lifting 10kg, were significantly associated with time to return to work. While computer use (4hrs) remained significant (in both the presence and absence of type of job within the model). That means that the effects shown in the final model presented in the manuscript are a valid representation of the data and not an artefact of over-adjustment.

4. A related comment is the parsimony concept: with a number of subjects 201 and 19 variables in the cox model, I feel this is not the case. I would advise to select relevant variables before where proportional risk might also be confirm. Authors should stay on the variable that are supposed to be associated and that make sense.

The variables included in the participant questionnaires (and therefore candidate variables) were all factors that we believed might be associated with the duration of work absence. The study was designed to answer the research questions presented (rather than as a secondary analysis of an existing dataset). The selection of candidate variables is discussed in response to point 1 from reviewer 1 and additional information on the reasoning for the inclusion of each variable is provided as a supplementary file (supplementary file 1).

The final mutually adjusted model comprised those variables which were significant at the 5% level in the initial variable+age+sex adjusted analyses. To test the stability of the final model, we tested this by comparing with a model using a 1% cut-off. There were fewer variables in the model, but the findings remained, with similar effect sizes. This suggests that our model is an appropriate representation of the data. Please also see our response to reviewer 1, point 2.

The sample size may indeed be small for the number of variables used. However, it roughly corresponds to 10 events per variable in the model, which is commonly used as a rule of thumb for survival models. Additionally, if anything, low statistical power would result in accepting a false null hypothesis, which is not the case in our data since associations found were both significant and plausible.

5. NB In table 4, CTS6 = Poor (8.8-5) should be Poor (3.8-5)

Thank you for spotting this. Apologies that the error crept into the original manuscript. We have made this correction.

VERSION 2 – REVIEW

REVIEWER	Dr Claire Burton School of Medicine, Keele University, UK
REVIEW RETURNED	10-Dec-2020

GENERAL COMMENTS	Thank you for taking the time to answer the points raised with depth and clarity. A few issues remain, which require further clarification / acknowledgement in the discussion limitations. 1. My concern regarding the number of candidate prognostic factors considered in the model development remains. Riley et al 2020 : https://www.bmj.com/content/368/bmj.m441.full.print may provide helpful information. 2. Can you explain why missing data was not imputed and coded as a separate category? Were data considered to be missing not at random? If not, this needs to be discussed as a potential limitation. 3. Given the model may be underpowered, consider a method of shrinkage to minimise the risk of overfitting. It would then be helpful
--

	to present the model equations and provide measures of performance (calibration and discrimination). Internal validation could then be carried out using a procedure such as boot-strapping. If the assumptions of the cox model have been tested, the results also need reporting. If this can not be done due to the original SAP - please consider discussing these as limitations. Steyerberg et al 2013 https://journals.plos.org/plosmedicine/article?id=10.1371/journal.pmed.1001381 and references may be helpful. 4. I'm not sure that GP's are performing CTR in every case of community CTR - the services commissioned to run in a primary care setting are often performed by hand surgeons / consultant nurses. It may be more accurate to discuss as CTR in the primary care (or community) setting. 5. Ethical permissions are assumed but not stated in the text. I note the IRAS number on the questionnaires / PIS and consent forms.
--	---

VERSION 2 – AUTHOR RESPONSE

REVIEWER 3

Comments to the Author

The authors have considered all comments and the manuscript is clear

Authors' response

Thank you for this feedback and for taking the time to review our work.

REVIEWER 1

Point 1.

Thank you for taking the time to answer the points raised with depth and clarity. A few issues remain, which require further clarification / acknowledgement in the discussion limitations.

1. My concern regarding the number of candidate prognostic factors considered in the model development remains. Riley et al 2020 : <https://www.bmj.com/content/368/bmj.m441.full.print> may provide helpful information.

Authors' response

Thank you for raising this point. We have read the recommended papers and they share some interesting insights for the development of prediction models. Our study did not aim to develop a prediction model, rather it was an attempt to explore the association between potential risk factors in relation to the duration of work absence after CTR. We were not looking to predict the duration of work absence for any future patients, rather explore potential risk factors and highlight areas for future research. We have added additional statements within the discussion to ensure that this is clear for the reader.

Amended section: Discussion (page 16)

Additional text: The reported model does not look to predict the duration of work absence for future CTR patients, rather to explore the association between different risk factors and work absence.

Amended section: Discussion (page 19)

Additional text: Furthermore, we acknowledge that the inclusion of a large number of variables in the

development of the final model may result in model overfitting, thereby potentially limiting generalisability.

Point 2

2. Can you explain why missing data was not imputed and coded as a separate category? Were data considered to be missing not at random? If not, this needs to be discussed as a potential limitation.

Authors' response

Thank you for highlighting this potential limitation. Missing data were coded as 'missing' part of our a priori analysis plan. This is documented in the statistical methods section. To help clarify this we have now amended Table 4 to show the number missing for each variable, and the 5 censored individuals (that is the 5 individuals who had not returned to work and therefore did not contribute to the median return to work time, but were included in the Cox proportional hazards model, censored to the last point of data collection). In response to this specific comment we have now also updated this information in the supplementary data file to ensure transparency.

Overall, the amount of missing data was very small and at the item-level, rather than across multiple variables. We are therefore confident that this low level of missing data supports the validity of the model, and that we now transparently report missing data in the manuscript. To provide further reassurance to the reader we have also added a statement within the limitations section acknowledging the amount and possible impact of missing data.

Amended section: Table 4, supplementary file 4 and discussion (page 20)

Additional text: We took the approach not to impute values where data were missing. Overall, the amount of missing data was small and at the individual item-level (Table 4 and supplementary file 4). Missing data were coded as such, and included in the analysis. We acknowledge that the approach taken to missing data may have resulted in biased estimates, yet if such effects are present, they are likely to be minimal due to low levels of missing data.

Point 3

3. Given the model may be underpowered, consider a method of shrinkage to minimise the risk of overfitting. It would then be helpful to present the model equations and provide measures of performance (calibration and discrimination). Internal validation could then be carried out using a procedure such as boot-strapping. If the assumptions of the cox model have been tested, the results also need reporting. If this can not be done due to the original SAP - please consider discussing these as limitations. Steyerberg et al 2013

<https://journals.plos.org/plosmedicine/article?id=10.1371/journal.pmed.1001381> and references may be helpful.

Authors' response

As part of our assessment that the assumptions proportional hazards were met, we visualised Kaplan-Meier curves. These were non-overlapping and roughly parallel. We also assessed the Schoenfeld residuals, with a resulting global test p value (>0.05) suggesting that the null hypothesis of proportional hazards should be accepted. There were no time-dependent covariates. We have included the assessment of proportionality in the manuscript.

We appreciate that our final model may be underpowered. We have demonstrated that our model appears robust with the collected data, and have acknowledged its limitations. We have been transparent that our work is based on a convenience sample of patients, and the aim is to explore all factors that have been previously reported as potentially associated with the duration of work absence. The aim of our research was not to develop a model to predict duration of work absence after CTR. Our hope is that our findings promote interest in return to work as an important outcome after elective hand surgery and inspire additional high-quality hand surgery research based on our exploratory work.

Amended section: Statistical methods (page 8)

Amended text: A cox proportional hazards model was used to explore the factors associated with return to work time, and assumptions of the model were tested.

Point 4

4. I'm not sure that GP's are performing CTR in every case of community CTR - the services commissioned to run in a primary care setting are often performed by hand surgeons / consultant nurses. It may be more accurate to discuss as CTR in the primary care (or community) setting.

Authors' response

Thank you for highlighting this. In the current study, all cases of CTR in primary care were performed by GPs. These individuals were part of the Association of Surgery in Primary Care (ASPC) and had received additional training through the ASPC. The lead author recruited participating primary care sites through the ASPC. We are aware of one site in the UK where an Advanced Clinical Practice nurse performs CTR, the lead author met with this individual during the development stage of this study, but they were not involved in the study delivery. They did not know of any other cases of Advanced Clinical Practice nurse surgeons providing CTR in the UK. We acknowledge that not all cases of CTR in primary care will be performed by GPs and have added this as an additional limitation in terms of generalisability.

Amended section: Discussion (page 17)

Amended text: We acknowledge that hand surgeons may also provide CTR services in primary care, as visiting clinicians, however in the current study this was not the case. CTR and other surgical procedures, such as vasectomy and minor skin surgery, are regularly performed by trained general practitioners in the UK[27], and all primary care surgeons in the current study were general practitioners who already provided a CTR service.

Amended section: Discussion (page 20)

Amended text: The findings may not be generalisable to working populations in regions outside of central and southern England and Wales, who are employed in other industries, or managed with a different CTR patient pathway. Steps were taken to include the main settings where CTR is performed in the UK, but we acknowledge that CTR may also be performed by other specialities.

Point 5

5. Ethical permissions are assumed but not stated in the text. I note the IRAS number on the questionnaires / PIS and consent forms.

Authors' response

Thank you for this suggestion. The 'ethics approvals' section was provided at the end of the manuscript, however to clarify this for the reader, we have also added the NHS ethics approval details in the main methods section.

Amended section: methods (page 5)

Amended text: (ethics approval: IRAS 209840, 16/WA/0390).

VERSION 3 – REVIEW

REVIEWER	Dr Claire Burton Keele University, United Kingdom
REVIEW RETURNED	12-Jan-2021
GENERAL COMMENTS	Thank you for your detailed responses. I think the additions made

	are helpful and provide clarification and acknowledgment of some of the potential limitations of the analysis. I do feel it would be good practice to provide a measure of the predictive power of the model (R squared, for example) and do not feel that the added text in line 55 of page 16 is particularly helpful. "The reported model has not been developed or validated to predict the duration of work absence for future CTR patients" may more descriptive.
--	--

VERSION 3 – AUTHOR RESPONSE

Reviewer: 1

Comments to the Author:

Thank you for your detailed responses. I think the additions made are helpful and provide clarification and acknowledgment of some of the potential limitations of the analysis.

I do feel it would be good practice to provide a measure of the predictive power of the model (R squared, for example) and do not feel that the added text in line 55 of page 16 is particularly helpful.

"The reported model has not been developed or validated to predict the duration of work absence for future CTR patients" may more descriptive.

Authors' response:

Thank you for taking the time to review our amendments and for your helpful suggestions.

1. Following your recommendation, we have calculated the Rsquared value for our model and present this within the text.

Amended section: Results, page 12

Additional text: The assessment of R2 indicated that 46% of variation in the duration of work absence was explained by the model (R2=0.46, 95% CI 0.37-0.53).

Amended section: Discussion, page 21

Additional text: Furthermore, we acknowledge that our model explained only 46% of variation in the duration of work absence.

2. We have also revised the suggested section of text within the discussion (previously line 55, page 16).

Amended section: Discussion, page 16

Amended text: The reported model has not been developed to predict the duration of work absence for future CTR patients, rather to explore and identify important risk factors for consideration in future research.